# Non-Invasive Ultrasonic Description of Tumor Evolution

**DOI:** 10.3390/cancers13184560

**Published:** 2021-09-11

**Authors:** Jerome Griffon, Delphine Buffello, Alain Giron, S. Lori Bridal, Michele Lamuraglia

**Affiliations:** 1Sorbonne Université, CNRS, INSERM, Laboratoire d’Imagerie Biomédicale, LIB, F-75006 Paris, France; jerome.jn.griffon@gmail.com (J.G.); delphine.le_guillou@sorbonne-universite.fr (D.B.); alain.giron@sorbonne-universite.fr (A.G.); lori.bridal@sorbonne-universite.fr (S.L.B.); 2AP-HP, Hôpital Beaujon, Service d’Oncologie Digestive et Medicale, F-92110 Clichy, France

**Keywords:** tumor microenvironment, CEUS, SWE, histological biomarkers, tumor neo angiogenesis, tumor growth

## Abstract

**Simple Summary:**

During tumor evolution, heterogeneous structural and functional changes occur in the tumor microenvironment. These complex changes have pro- or anti-tumorigenesis effects and have an impact on therapy efficiency. Therefore, the tumor microenvironment needs to be non-invasively characterized over time. The aim of this preclinical work is to compare the sensitivity of modifications occurring during tumor evolution of volume, immunohistochemistry and non-invasive quantitative ultrasound parameters (Shear Wave Elastography and dynamic Contrast-Enhanced Ultrasound) and to study the link between them. The complementary evaluation over time of multiple morphological and functional parameters during tumor growth underlines the need to integrate histological, morphological, functional, and, ultimately, genomic information into models that can consider the temporal and spatial variability of features to better understand tumor evolution.

**Abstract:**

Purpose: There is a clinical need to better non-invasively characterize the tumor microenvironment in order to reveal evidence of early tumor response to therapy and to better understand therapeutic response. The goals of this work are first to compare the sensitivity to modifications occurring during tumor growth for measurements of tumor volume, immunohistochemistry parameters, and emerging ultrasound parameters (Shear Wave Elastography (SWE) and dynamic Contrast-Enhanced Ultrasound (CEUS)), and secondly, to study the link between the different parameters. Methods: Five different groups of 9 to 10 BALB/c female mice with subcutaneous CT26 tumors were imaged using B-mode morphological imaging, SWE, and CEUS at different dates. Whole-slice immunohistological data stained for the nuclei, T lymphocytes, apoptosis, and vascular endothelium from these tumors were analyzed. Results: Tumor volume and three CEUS parameters (Time to Peak, Wash-In Rate, and Wash-Out Rate) significantly changed over time. The immunohistological parameters, CEUS parameters, and SWE parameters showed intracorrelation. Four immunohistological parameters (the number of T lymphocytes per mm^2^ and its standard deviation, the percentage area of apoptosis, and the colocalization of apoptosis and vascular endothelium) were correlated with the CEUS parameters (Time to Peak, Wash-In Rate, Wash-Out Rate, and Mean Transit Time). The SWE parameters were not correlated with the CEUS parameters nor with the immunohistological parameters. Conclusions: US imaging can provide additional information on tumoral changes. This could help to better explore the effect of therapies on tumor evolution, by studying the evolution of the parameters over time and by studying their correlations.

## 1. Introduction

The composition of the tumor microenvironment changes heterogeneously over time. During tumor growth, a neo-vascular network develops in the tumor. This neo-angiogenesis helps the tumor to grow [1], but this new vascular network is anarchic; it becomes more tortuous and leaky as the tumor grows. Non-vascular zones develop and the vascular network becomes less efficient [2,3,4]. Apoptosis of cancer cells also impacts angiogenic development. Studies have shown a relationship between apoptosis reduction and tumor growth, due to the release of neo-angiogenic factors [5]. Moreover, the vascular network can play an important role in immune response [5]. It was shown that T cells are dependent on normal oxygen levels for migration into the tumor tissue [6]. This suggests that hypoxia indirectly fetters T-cell access [7,8]. Thus, the state of the vascular network can have a pro- or anti-tumorigenesis effect. The microenvironment changes also involve the extracellular matrix. Erler and Weaver in 2009 demonstrated that the synthesis and remodeling of fibrillar type I collagen increases in tumors and is required for angiogenesis [9]. The anarchic and progressive changes of fibers promote the metastatic process [10,11]. Finally, because of the remodeling of the extracellular matrix, tumors are often stiffer than the surrounding normal tissue [7,8].

All these complex structural changes have consequences on therapy efficiency, making it difficult to do a conclusive and early analysis of treatment response. The only currently validated clinical indicator used to monitor tumor evolution during therapy follow-up is the Response Evaluation Criteria in Solid Tumors (RECIST) 1.1, based on tumor morphology [12,13]. This criterion does not take into account the complexity of the structural and functional changes in the tumor microenvironment during tumor growth. Thereby, it does not give direct information on the microenvironment or the effect that a therapy can have on the microenvironment. Therefore, this index can be the expression of a late-stage microenvironment modification in vivo [14,15]. In spite of these limitations, RECIST 1.1 is the only standardized criteria. It is reproducible and independent of image-system constructers and operators. In the literature, some authors have combined morphological and functional criteria: mRECIST criteria (Modified Response Evaluation Criteria in Solid Tumors) [16], Choi and mChoi criteria [17,18], and iRECIST criteria (immune Response Evaluation Criteria in Solid Tumors) [19]. However, although these criteria have shown interesting results, their use remains limited due to concerns about assessment reproducibility and a lack of standardization.

Histological analyses can give partial information about the tumor microenvironment. However, this method is invasive and only samples a very limited part of the whole tumor [20]. There is, therefore, a clinical need to better non-invasively and more globally characterize the tumor microenvironment. The observation of one or multiple biological characteristics of the tumor using biomarkers obtained with imaging techniques can reveal evidence of early tumor response to therapy and can help, in the end, to better understand therapeutic response. This could notably be used during clinical tests of new therapies to evaluate efficiency [14]. Ultrasound imaging techniques have shown their capacity to monitor therapeutic effects that occur earlier or that are more subtle than the tumor size reduction assessed by RECIST [21,22,23,24,25].

Ultrasound Elastography can estimate the mechanical properties of tissue by the measurement of their elasticity. These techniques are based on the measurement of the displacement of the tissue induced by a mechanical perturbation [26]. This measure is then linked to the local elastic properties of the tissue. Among the existing Ultrasound Elastography imaging techniques, Shear Wave Elastography (SWE) provides a quantitative evaluation based on the local speed of propagation of a shear wave produced by the probe [26]. It has been shown that the elasticity estimated by SWE was linked to the presence of necrosis and fibrosis in a xenograft tumor model of human HBCx-3 [27].

Dynamic Contrast-Enhanced Ultrasound (CEUS) uses intravenously injected microbubbles as a contrast agent. This technique can help to characterize the state of the vascular network in the tumor. Contrast agent microbubbles can be targeted to a specific receptor of blood vessels, such as the Vascular Endothelial Growth Factor (VEGF) with BR55 [28]. Other contrast agents, such as the SonoVue [29], freely circulate in blood vessels.

Each ultrasound imaging technique has been studied on different tumor models and compared to histological data [27,30,31,32]. This work presents a tumor evolution monitoring approach that considers all these imaging techniques (whole-slice histological analysis and non-invasive quantitative ultrasound SWE and CEUS) at different time points in one experiment. This allows a comparison of the structural and functional parameters of a heterogeneous tumor during its complex evolution over time. In this work, the sensitivity to modifications occurring during tumor growth are compared for the reference parameters (tumor volume and immunohistochemistry) and emerging ultrasound parameters (SWE and CEUS), and the link between the different parameters is studied. The evolution over time and correlation between the mean values of the different parameters in the tumor are evaluated.

## 2. Materials and Methods

### 2.1. Experiment

Initially, a 100 μL suspension of colorectal carcinoma CT26 cells at a concentration of $2\times 10^{6}$ cells/mL were injected subcutaneously into the flank of 3 BALB/c mice. The mice were euthanized 21 days later and fragments of the CT26 tumors with the major axis from 2 to 3 mm were subcutaneously implanted into the left flank of 49 BALB/c female mice (Janvier Labs, St. Berthevin, France). In previous studies of this ectopic murine colorectal carcinoma model, it has been shown that tumor fragments are effectively engrafted around 10 days after implantation based on both histological analysis of the microvascular density [30] and on the angiogenesis stage starting around this day, as observed using T2-weighted MRI images [33]. Therefore, in this model, analysis of tumor evolution and correlations between the parameters should be considered after the tumor has had time to reach this stage (Day 9 (D9) in our measurements).

At five different dates after implantation (D7, *n* = 10; D10, *n* = 10; D14, *n* = 10; D16, *n* = 9; and D17, *n* = 10), groups of mice were imaged using B-mode morphological imaging, SWE, and CEUS. The mice examined each day were chosen so that the mean and the standard deviation of the volume distribution of the remaining population were similar before and after the removal of the imaged mice. To achieve this, 2 to 3 mice were chosen in each quartile of the tumor volume distribution of the total population. As explained above, data at D7 will not be analyzed. 

For each imaging technique, the transducer was positioned in a fixed support and acoustic coupling was made through a thick layer of echographic gel (around 1 cm thick). Use of a support minimized pressure on the skin surface and avoided operator-related movements. After being imaged, the mice were euthanized, and the tumors were excised and marked to conserve the orientation and approximate position relative to the US imaging plane. Immediately after excision, tumors were immersed in a glucose solution for 24 h. Tumors were then frozen with liquid nitrogen in an Optimal Cutting Temperature compound cube and stored at −80 °C. Out of the 49 samples, 30 exploitable tumors were analyzed using immunohistological techniques (D7, *n* = 1/10; D10, *n* = 6/10; D14, *n* = 8/10; D16, *n* = 7/9, and D17, *n* = 8/10). The protocol was approved by the Charles Darwin ethics committee (Ref# 1837 2015092214507792).

### 2.2. Tumor Volume

Using an Aixplorer (SuperSonic Imagine, Aix en Provence, France) ultrasound imaging system with an SL15-4 probe in B-mode, the width and the thickness of the largest section of the tumor along the transversal and the longitudinal planes were measured perpendicular to the head–tail axis and along this axis, respectively. The tumor volume was then estimated using the ellipsoidal approximation $\;V=\frac{\pi }{6}*abc$, where *a* represents the width of the tumor along the transversal plan, *b* represents the width of the tumor along the longitudinal plan, and $c=\frac{c_{1}+c_{2}}{2}$, the mean between the measured thickness in each plane. According to Cheung et al., an ellipsoidal approximation is acceptable for estimating the tumor volume [34].

### 2.3. SWE

Tissue elasticity distribution maps (ShearWave Elastography-SWE) were acquired and extracted as DICOM images with the same device and probe (SSI, Aixplorer, SL15-4 probe) at the major longitudinal plane using the penetration mode and a medium time persistence. The SWE window was centered on the tumor with the edges extending well beyond the tumor boundaries and the focus positioned near the center of the tumor. The SWE map displayed by the ultrasound system presented visible changes over time for approximately 15 s. Then, the operator chose the image to analyze as the most stable image over time having the least non-filled zones (zones without valid SWE estimations). Since the breathing rate of the mice under anesthesia is low and stable, we were able to choose an image where the mice were in a motion-free state of their breathing cycle. Quantitative maps were extracted for each image using custom software developed by Supersonic Imagine (Aix-en-Provence, France). The average and standard deviation of SWE were calculated within regions outlined along the tumor boundary on the corresponding B-mode image, excluding artifacts. 

### 2.4. CEUS

The imaging plane for dynamic Contrast-Enhanced Ultrasound (CEUS) (Sequoia 512, 15L8w-S probe 7–14 MHz, contrast pulse sequencing cadence (CPS)) was positioned while observing the B-mode image of the longitudinal plane acquired with the Aixplorer system used to acquire the SWE data (examples of the comparison of the two B-mode images can be seen in Figure 1). The dynamic range was set to 80 dB and the CPS gain to 0 dB. The mechanical index was set at 0.1 to minimize destruction of the microbubbles. The focus was placed in the distal region of the tumor. A bolus injection of SonoVue (Bracco Suisse, Geneva, Switzerland) was made in the caudal vein with a controlled injection system [35]. Each mouse received an injection of a solution made of 20 mL of SonoVue and 30 mL of physiological serum. This corresponds to a quantity computed for a mouse of 20 g with a concentration of SonoVue of 1 mL/kg. The 50 mL solution was injected using a flush of 0.100 mL of physiological serum at a speed of 4.5 mL/min made by a syringe pump (Pump11, Harvard Apparatus, Holliston, MA, USA). The imaging acquisition began just prior to the injection. The frame rate was set at 3 images per second during the first 30 s after detection of microbubbles in the imaged plane and then reduced to 1 image per second for the rest of the sequence (2 to 3 min long). The extracted DICOM CEUS sequences were manually segmented and the echo-power was then estimated [36]. A lognormal model recommended by the EFSUMB [37] was fit to the time–intensity curve obtained from inside the tumor by minimization of the mean square error criterion using a nonlinear optimization method. From this modelling, the parameters related to microvascular volume and flow were extracted [31,38]. The estimated parameters were the Area Under the Curve (AUC), Peak Enhancement (PE), Time to Peak (TTP), Mean Transit Time (MTT), Wash-In Rate (WIR), and Wash-Out Rate (WOR).

### 2.5. Immunohistology

Slices of the tumor, 10-µm thick, were sectioned with a cryostat (CM3050, Leica, Nussloch, Germany) along the largest cross-section aligned according the marks made during excision to conserve the orientation with respect to the imaged plane. Whole-slice histological sections were then prepared with fluorescent immunohistochemical markers for reference assessment of the cell nuclei (DAPI), T lymphocytes (antibody CD3-Alexa 647), apoptosis (antibody anti-caspases 3 (BioVision, ref: 3015-100)—TRITC), and vascular endothelium (antibody isolectine B4—Alexa 488). These marked slices were then scanned (Axio Scan Z1, ZEISS, Jena, Germany) at a resolution of 0.325 × 0.325 μm. Whole-slice histological section images were manually segmented to define the outer boundaries of each tumor. The regions of each histological section that were free of artifacts (localized tears and folds) were analyzed using in-house software to estimate in the whole slice the percent area occupied by nuclei (NU), the number of T Lymphocytes per mm² (NTL), the percent area of apoptosis (AP), and the percent area of vascular endothelium (VE) marker. Parametric maps were made by computing these parameters in regions of 512 × 512 pixels (0.166 × 0.166 mm). The standard deviation of the parameters in each slice was computed from the parametric maps. High AP regions in the parametric maps were defined where the AP of the pixels was above 3%. High VE regions were defined where the VE of the pixels were among the highest 15% for the analyzed section. Based on these thresholded maps, the % surface with co-localization of high VE and AP (VE ∩ AP) was estimated.

### 2.6. Statistical Tests

Because of the limited number of samples in the study and the independence of the data, the difference between the days was tested using the non-parametric ranked Kruskal–Wallis test followed by a post hoc Steel–Dwass analysis. The test was made using the software JMP (JMP Pro Version 14.3.0, SAS Institute Inc., Cary, NC, USA, 1989–2019). The difference was significant if the *p*-value was < 0.05. The correlations between the parameters were calculated using the non-parametric Spearman test (correlation coefficient $r_{S}$) with MATLAB (MATLAB and Statistics Toolbox Release 2016b, The MathWorks, Inc., Natick, MA, United States). A correlation was significant when the associated *p*-value was <0.05. The correlations were computed using the data from all days, excluding data from D7. Correlation tables were built to visualize the correlation between the parameters and only the significantly correlated ones are shown.

## 3. Results

### 3.1. Evolution over Time

The evolution of the median and interquartile range of all the parameters over time can be seen in Table 1.

#### 3.1.1. Tumor Volume and Immunohistological Parameters

Tumor volume was significantly different between D10 vs. D14, D16, and D17 (Figure 2a). The mean and standard deviation of the immunohistological parameters representing the density of the biological structures did not show significant differences over time, except for the colocalization of apoptosis and the vascular endothelium (VE $\cap^{\;}\;$ AP). For this parameter, there was a significant difference between D10 and D14 (Figure 2c). The difference between days for this parameter can be noticed from an example of the parametric maps (Figure 3).

#### 3.1.2. Ultrasound Parameters

There was no difference over time for the mean and the standard deviation of SWE in the tumor. An example of the SWE parametric maps at different days are presented in Figure 3. For the CEUS imaging technique, the parameters WIR and WOR showed a significant difference between D10 and D17 (Figure 2d,e). The parameter TTP showed a significant difference between D10 and D14, D16, and D17 (Figure 2b). The other parameters (AUC, MTT, and PE) did not show significant differences between the days.

### 3.2. Correlation

#### 3.2.1. Correlation between Tumor Volume and Histological Parameters

The only immunohistological parameter that significantly correlated with the tumor volume was the standard deviation of percent area of vascular endothelium ($r_{S}=-0.41$). This negative correlation shows that the tumors tend to have a lower percent vascularization as they grow. Immunohistological parameters showed some intercorrelation. The percentage area of nuclei, vascular endothelium, and number of T lymphocytes per mm^2^ were positively correlated. The percent area of vascular endothelium and number of T lymphocytes per mm^2^ and their respective standard deviation formed another group of positive correlations. The third group of positive correlations was composed of the percent area of apoptosis and the standard deviation of percent area of nuclei, vascular endothelium, and apoptosis. Finally, the standard deviation of the nuclei and apoptosis and the percent area of apoptosis and the colocalization of vascular endothelium and apoptosis were positively correlated. The positive correlation between the percent area of apoptosis and the number of T lymphocytes per mm^2^ was also present. The strongest correlations were between VE and NTL ($r_{S}$ = 0.78), VE and σVE ($r_{S}$ = 0.77), AP and σAP ($r_{S}$ = 0.94), and AP and VE $\cap^{\;}\;$ AP ($r_{S}$ = 0.81) (Figure 4).

#### 3.2.2. Correlation between US Parameters

Groups of correlations (AUC, PE, WIR, and WOR) and (PE, WIR, WOR, and TTP) are present. There are two additional correlations between MTT and AUC and between MTT and TTP. Concerning the SWE parameters, the mean of the SWE and standard deviation of SWE in the tumor are positively correlated (Figure 4). There is no correlation between the SWE and CEUS parameters.

#### 3.2.3. Correlation between Volume and US Parameters

The significant correlations between the imaged parameters and the tumor volume are shown in Figure 4. The significant correlations were mainly moderate or weak ($r_{S}\leq 0.5$). The elasticity standard deviation (σSWE) was weakly and positively correlated to tumor volume. The mean of elasticity ($\bar{M}SWE$) did not correlate with tumor volume (Figure 4).

#### 3.2.4. Correlation between Immunohistological and US Parameters

Four immunohistological parameters were correlated with the CEUS parameters (Figure 4). The number of T lymphocytes per mm^2^ was correlated with TTP ($r_{S}=-0.45$), MTT ($r_{S}=-0.43$), WOR ($r_{S}=-0.42$), and WIR ($r_{S}=+0.39$). The standard deviation of the number of T lymphocytes per mm^2^ was correlated with TTP ($r_{S}=-0.38$) and WOR ($r_{S}=-0.37$). Finally, the percentage area of apoptosis and the colocalization of apoptosis and vascular endothelium (VE $\cap^{\;}\;$ AP) were correlated with TTP ($r_{S}=-0.44$ and $r_{S}=-0.40$, respectively). There was no correlation between the immunohistological parameters and the SWE parameters (Figure 4).

## 4. Discussion

Since this tumor model has been previously characterized by others, we can confirm that the measurements are consistent overall with previous observations. We observed a growth of the tumor volume over time as expected and reported in other studies on CT26 ectopic implantations [33]. The relative stability of the immunohistological parameters over time is coherent with other studies carried out on the same tumor model. For example, the microvascular density and the cellularity visually analyzed in the viable rim of histological slices was reported to be stable during tumor growth between D10 and D18 after subcutaneous implantation of the CT26 fragments [33]. The quantitative analysis of 10 sub-regions in the histological slices made by Seguin et al. in the same tumor model show a similar stability for microvascular density and cellularity [30]. The lack of change over time for the mean elasticity in the tumor was also observed in the study of Jugé et al. [33]. At different times during tumor growth, they measured the mean elasticity in viable zones of the tumor using an MRI elastography sequence. This mean was stable between the eleventh day after implantation until the end of the experiment (eighteen days after implantation). The CEUS parameters AUC, PE, and MTT did not show significant differences over time in our study. Turco et al. studied the CT26 model using a targeted CEUS imaging technique [32]. The three CEUS parameters that they considered can be related to the AUC, PE, and MTT parameters from the lognormal model. For the control group receiving intravenous injection of sterile saline solution, they did not observe significant differences during 10 days for any of the measured CEUS parameters. Measurements started 10 days after subcutaneous injection of tumor cells. The lack of variation that they observed is therefore coherent with the experiments we carried out. 

In our study, the standard deviation of the percent area of vascular endothelium was negatively correlated with the tumor volume. This would suggest that the bigger the tumor is, the less variable the vessel concentration. Homogeneous repartition of vessels in tumors with the same tumor model has been visually observed by Seguin et al. [30]. This means that the vascular network is developed throughout the tumor, but part of this network can be non functional (Figure 3). The more the tumor increases in volume, the more its vascular network grows and spreads capillaries. However, its relative proportion in the tumor does not drastically change during tumor evolution, as suggested by the non-significant evolution over time of percentage area of vascular endothelium.

The significant positive correlation between the standard deviation of elasticity and tumor volume suggested that the bigger the tumor is the more the elasticity values are dispersed in the tumor. Thus, the spatial heterogeneity of elasticity is more important in larger tumors. The positive correlation between the mean and the standard deviation of elasticity in the tumor means that a tumor having a higher mean elasticity should present a higher variability of elasticity (Figure 3). This is coherent with the work of Chamming’s et al., where the authors observed an increase in the value of the standard deviation for measured elasticity in breast cancer xenografts during tumor growth [27]. 

The majority of CEUS parameters, except for AUC and PE, correlated with tumor volume. The sign of the correlations can be explained by the evolution of the vascular network during tumor growth, as described in several studies [2,3,4,25]. According to these studies, the tortuosity and leakiness in the vascular network increased during tumor growth. Nonfunctional zones appear in the network and it becomes less effective (Figure 3). This, however, does not mean that the tumor is less aggressive or develops more slowly. 

There is no correlation between the SWE and the CEUS parameters. This could be explained by the fact that the consistency of the tumor is a consequence of tumor inflammation. Admittedly, this inflammation increases the development of the vascular system, with an increase in TAMs and other stromal cells, such as endothelial cells, smooth muscle cells, fibroblasts, that secrete several angiogenic growth factors and proteinases [39]. Yet, all these micro-biological modifications did not show specific functional change with our CEUS parameters. The markers of immune activity and apoptosis are weakly related to the hemodynamic parameters in the tumor (Figure 4). However, there is no correlation between the vascular endothelium markers and the CEUS parameters. This absence of a correlation could be explained by the fact that the histological marker stains both the functional and nonfunctional vessels whereas CEUS imaging only shows functional vessels. We did not observe a correlation between the SWE parameters and immunohistological parameters. Jugé et al. [33] in their study on a CT26 model showed that the elasticity measured with MRI is positively correlated with cellularity. The fact that they only analyzed the viable part of the tumor in their studies could explain this difference to our study. We did not carry out the same analysis because the imaging modality was not the same (US vs. MRI) and we cannot have information on the viable part of the tumor using ultrasound modalities (the CEUS discriminates vascular and non-vascular zones but does not show viable and non-viable). 

The question of how to best deal with the resolution difference between US imaging and histology has been asked in several studies [40,41,42], due to the fact that the scales used are different between these two technologies: in the order of µm for histology and in the order of mm for ultrasound. Moreover, the fact that ultrasound imaging planes have a certain thickness implies that we observe an accumulation of biological effects [41]. The question of the difference of resolution between the imaging and the histology asked by several studies [40,41,42] can be solved by the construction of 3D histological models made from the combination of successive slices of tissue, as exposed by Wildeboer et al. [43,44]. The modeling of physical phenomenon based on the histology, as realized by Mamou et al., could also help to get over the resolution problem [45]. Finally, O’Connor et al. asked the question if the imaging and histology could represent the same thing; that is, if the two techniques measure the same biological effect, but in two different ways, or if they measure different biological effects and are, therefore, complementary [42]. In this study, the histology is mainly representing structural characteristics of the tissues and cannot show functional activities in the tissue that are evaluated with CEUS imaging and the information is, thus, complementary. Although the imaging planes have been carefully chosen and oriented in our experiment, an identical superposition between the different US imaging techniques and immunohistochemistry imaging cannot be assured. CEUS in this study is sensitive to this sampling variability. The level of variability to expect was characterized by Streeter et al. who showed that two adjacent slices 800 µm away from each other showed a mean of around 10% of variability in tumors studied using a targeted CEUS [46]. Faced with tumor heterogeneity and the need to identify new parameters that can help describe and predict tumor evolution, Heng et al. highlighted the potential of associating genomic parameters with developments in the macro- and micro-environment [47,48].

## 5. Conclusions

First, we have analyzed how the reference parameters (volume and immunohistochemistry) and the emerging ultrasound imaging parameters (SWE and CEUS) changed over time. The fact that the structural immunohistochemical parameters showed little or no significant evolution over time is consistent with previous works characterizing this tumor model. This lack of microstructural modifications was reflected by the fact that the average SWE values did not change markedly during tumor growth. The overall volume and complementary functional US imaging parameters, however, revealed changes over time during tumor growth. The US imaging parameters thus reflected a structural (SWE) and functional (CEUS) tumor status, consistent with the biological evolution of this model. We considered the effect of tumor growth on imaging parameters on a global, whole tumor scale, given that we considered the mean value and standard deviation on the whole tumor section in the imaging plane. However, we indirectly observed that the heterogeneity increased through the evolution of the standard deviation of SWE during tumor growth. More local mapping of the SWE and CEUS changes during tumor evolution could be useful to probe modifications of the tumor heterogeneity. Such complementary evaluations over time during tumor therapy should help to better explore the effect of therapies on tumor evolution. In conclusion, evaluation of multiple morphological and functional parameters in this heterogeneous tumor model underline the need to integrate histological, morphological, functional, and, ultimately, genomic information into models that can consider the temporal and spatial variability of features to better understand tumor evolution. The scientific community is beginning to probe the nonlinear natural history of cancer evolution, the heterogeneity, and the numerous factors that modify the pathway of cells more deeply. To better understand the impact of these factors, integrated information from histological, morphological, functional, biological, molecular, and genetic markers will be essential.

## Figures and Tables

**Figure 1 cancers-13-04560-f001:**
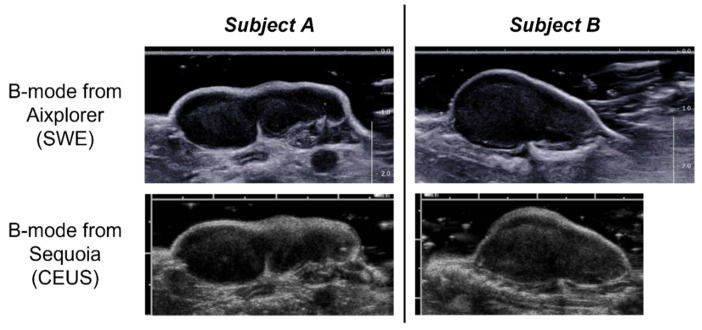
Example of the B-mode images along the longitudinal plane obtained with the Aixplorer ultrasound imaging system (used for ShearWave Elastography (SWE) imaging) and acquired at matched positions with the Sequoia system (used for dynamic Contrast-Enhanced Ultrasound (CEUS) imaging) for two different subjects. For the two imaging systems the distance between major tick marks is equal to 1 cm.

**Figure 2 cancers-13-04560-f002:**
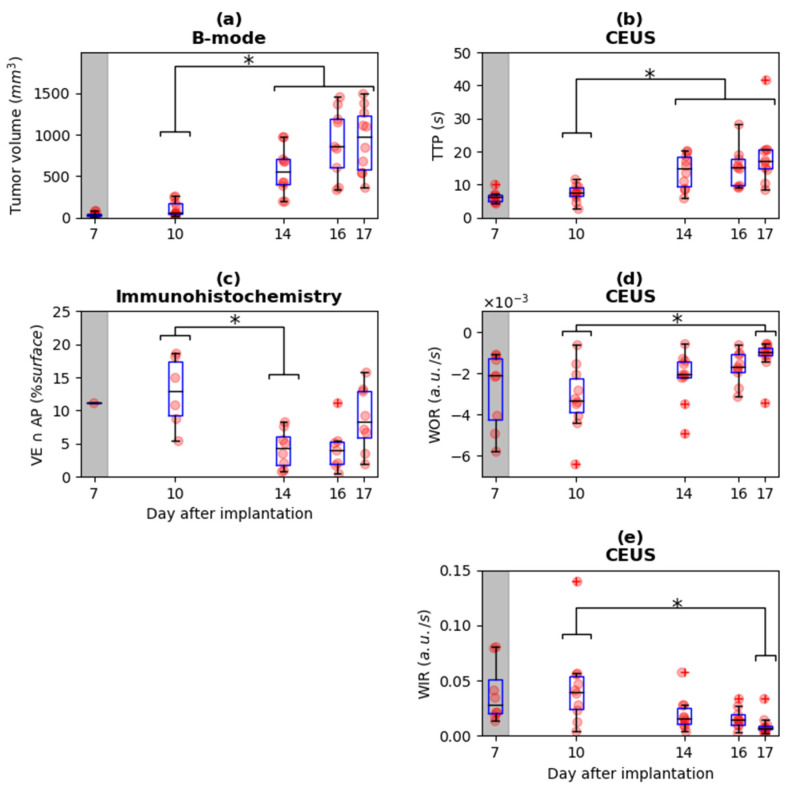
Evolution over time of (**a**) tumor volume, (**b**) Time To Peak (TTP), (**c**) colocalization of apoptosis and vascular endothelium (VE ∩ AP), (**d**) Wash-Out Rate (WOR), and (**e**) Wash-In Rate (WIR). For tumor volume (**a**) and the contrast parameters TTP (**b**), the values at D10 are significantly different to the values at D14, D16, and D17. The immunohistological parameter VE ∩ AP (**b**) shows significant difference between D10 and D14. The contrast parameters WOR (**d**) and WIR (**e**) show significant differences between D10 and D17. Initial measurements at D7 are shaded in grey because previous studies in this model suggest that the tumor fragment is not yet biologically integrated into the host prior to D9. The raw data and associated boxplot are overlaid; * = *p* < 0.05, + = outliers.

**Figure 3 cancers-13-04560-f003:**
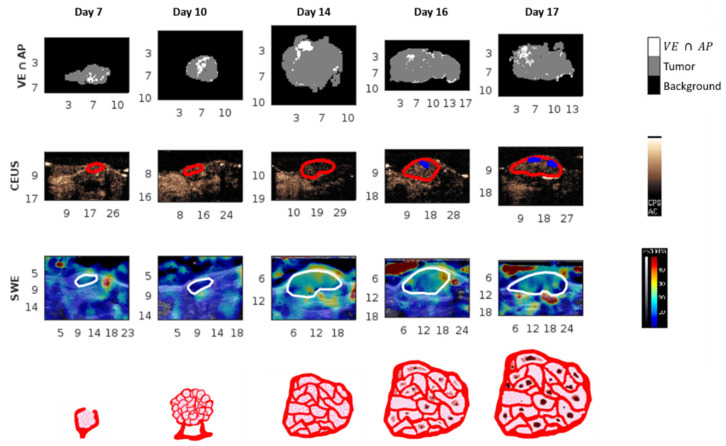
Example of a parametric map over time. Each column (day) shows data from the same tumor. Scales at the border of images are in millimeters. The first row is a representation of the immunohistological images of co-localization of a high percent area of the vascular endothelium and apoptosis (VE ∩ AP). The second row is a snapshot of the CEUS sequence with the tumors outlined in red and the nonperfused zones in blue. The third row is the SWE parametric map with the tumors outlined in white. The fourth row represents an illustration of the biological tumor state at the tumor cell and vascular level.

**Figure 4 cancers-13-04560-f004:**
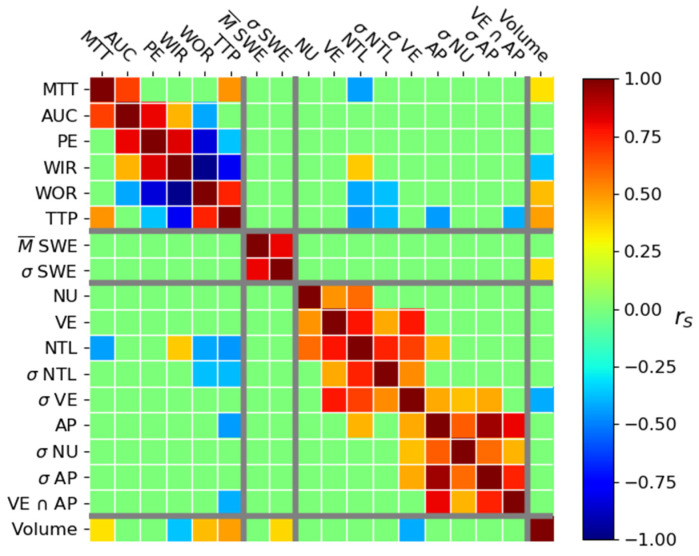
Spearman correlation coefficient value between the CEUS parameters (Mean Transit Time (MTT), Area Under the Curve (AUC), Peak Enhancement (PE), Wash-In Rate (WIR), Wash-Out Rate (WOR), and Time to Peak (TTP)), SWE parameters, histological parameters (percent area occupied by nuclei (NU), the number of T Lymphocytes per mm² (NTL), the percent area of apoptosis (AP), and percent area of vascular endothelium (VE), and co-localization of a high VE and AP (VE ∩ AP)), and tumor volume. Only significant correlations are shown (*p* < 0.05).

**Table 1 cancers-13-04560-t001:** Evolution of the median (interquartile range) of the following parameters over time: tumor volume (mm^3^), histological parameters (percent area occupied by nuclei (NU), the number of T Lymphocytes per mm² (NTL), the percent area of apoptosis (AP), and percent area of vascular endothelium (VE), and their spatial standard deviation *σ*, and the co-localization of a high VE and AP (VE ∩ AP)), the SWE parameters (mean $\bar{M}$ and standard deviation *σ* in kPa), and CEUS parameters (Area Under the Curve (AUC), Peak Enhancement (PE), Time to Peak (TTP), Mean Transit Time (MTT), Wash-In Rate (WIR), and Wash-Out Rate (WOR)).

Parameters	D7	D10	D14	D16	D17
Volume	28.3 (9.7)	64.4 (22.7)	554.4 (263.4)	862.6 (369.3)	978.1 (510.3)
NU	29.98 (−)	55.42 (7.77)	58.15 (13.36)	46.17 (0.99)	48.04 (3.83)
σ NU	19.71 (−)	18.57 (1.19)	17.55 (3.64)	20.15 (3.94)	17.79 (1.47)
NTL	861.18 (−)	597.38 (143.19)	636.09 (233.60)	282.78 (61.15)	321.02 (76.50)
σ NTL	2.88 × 10^3^ (−)	760.49 (148.61)	637.93 (289.35)	527.22 (129.09)	361.31 (105.64)
AP	2.29 (−)	2.36 (0.74)	1.00 (0.56)	0.80 (0.92)	1.63 (0.84)
σ AP	3.63 (−)	5.28 (2.26)	2.70 (0.56)	2.92 (1.95)	4.78 (2.06)
VE	27.32 (−)	38.49 (21.98)	37.48 (5.02)	20.18 (15.26)	32.93 (7.02)
σ VE	19.50 (−)	18.21 (0.91)	16.58 (2.72)	13.14 (4.57)	15.39 (2.50)
VE ∩ AP	11.24 (−)	12.97 (4.65)	4.36 (3.13)	4.05 (2.49)	8.25 (3.11)
$\bar{M}$ SWE	10.8 (3.3)	12.9 (4.5)	16.2 (3.9)	17.4 (2.9)	16.6 (2.2)
σ SWE	3.0 (1.1)	3.6 (0.4)	5.4 (1.3)	6.5 (1.7)	5.8 (0.6)
AUC	3.21 (0.84)	5.56 (2.32)	6.48 (3.35)	6.33 (1.29)	4.66 (2.52)
PE	7.3 × 10^−2^ (2.7 × 10^−2^)	0.12 (4.1 × 10^−2^)	9.2 × 10^−2^ (2.0 × 10^−2^)	9.6 × 10^−2^ (2.3 × 10^−2^)	6.0 × 10^−2^ (2.4 × 10^−2^)
TTP	6.23 (1.59)	7.63 (1.33)	14.77 (6.34)	15.37 (5.78)	17.19 (2.67)
MTT	68.53 (22.75)	64.58 (6.65)	68.90 (16.70)	76.22 (15.99)	67.10 (14.45)
WIR	2.8 × 10^−2^ (1.2 × 10^−2^)	4.0 × 10^−2^ (1.7 × 10^−2^)	1.6 × 10^−2^ (6 × 10^−3^)	1.5 × 10^−2^ (5 × 10^−3^)	7 × 10^−3^ (1 × 10^−3^)
WOR	−2 × 10^−3^ (2 × 10^−3^)	−3 × 10^−3^ (6 × 10^−4^)	−2 × 10^−3^ (1 × 10^−4^)	−2 × 10^−3^ (3 × 10^−4^)	−9 × 10^−4^ (2 × 10^−4^)

## Data Availability

The data presented in this study are available upon request from the corresponding author. The data are not publicly available due to regulations of the institution.

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
