# Peer review of "Non-Invasive Ultrasonic Description of Tumor Evolution"

_cancers, 2021, doi:10.3390/cancers13184560_

Round 1

Reviewer 1 Report

The authors have addressed some issues raised by the reviewer satisfactorily. There are still a lot of issues that need to be corrected before publication. The manuscript can be improved by considering the following comments:

  1. The main limitation of the study is that most of the parameters either from Histochemistry or imaging did not show a significant difference over time. Though the main objective was to find change in the tumor microenvironment over time, tumor volume change was better than other parameters. There was no good explanation about why Histochemistry parameters did not show any significant difference but volume showed. The authors tried to explain by citing Juge et al. 2012, but Juge et al. had an earlier time point. The authors should add earlier time points in the study. Showing correlation is not enough. Because correlation does not mean causation. For example, mean SWE is correlated with volume change and standard deviation of SWE. However, the standard deviation of SWE is not correlated with volume change.
  2. The authors should consider Ultrasound B-mode imaging, Shear wave elastography, and CEUS as 3 separate imaging modalities instead of lumping them under overall ultrasound imaging parameters.
  3. Innovation is still not clear in the introduction. Just mentioning that we want to study the correlation between the US versus Histochemistry parameters is not enough. As mentioned previously, the investigation of the relationship between Histochemistry versus SWE parameters and versus CEUS parameters was done previously in both human or mouse models (example, Ref. 27 and Payen et al. UMB, 2015 Aug;41(8):2202-11). The authors need to clear the innovation in the introduction by citing previous works related to the relationship between Histochemistry versus US parameters and how their work is different from the previous work.
  4. The authors need to show the non-significant parameters over time in a table as mean ± standard deviation.
  5. The authors should show B-mode images from Sequoia and Axiplorer systems and how B-mode images of two different systems matched with each other.
  6. The authors need to provide a cartoon or sketch of the experimental setup because the cartoon may help others to design the experimental setup and reproduce the results.
  7. Fig 1: The authors should add a title to each panel to remind the reader which parameters come from which imaging methods. Panel (c), Why did the “VE n AP” increase at D17?
  8. Fig 2: The authors need to significantly improve the Fig. 2
    • The authors should add B-mode images at each time point. As the authors are calculating tumor volume and boundary from the B-mode images, It will help the reader to compare B-mode versus other imaging modalities.
    • The authors should add Histochemistry images of VE and AP separately at each time point. Without the separate image of VE and AP, the image of “VE n AP” does not make sense.
    • The authors should also add other histochemistry images like DAPI and CD3.
    • Instead of showing CEUS images, the authors should show the image of different CEUS parameters.
    • How did the authors derive row 4 images showing the biological state of tumors?
  9. Fig. 3: The authors should only show one side of the main diagonal to make the figure less complex. Correlation with p>0.05 should be represented by a different color like black or white.
  10. Abstract, Results: The author should provide quantitative values for correlation instead of just mentioning correlated.
  11. Introduction, Line 57: Define RECIST and what is the criteria?
  12. Was SWE assessment performed by breathing gated? If not, how does the breathing of mice impact SWE measurements?
  13. Page 3, Tumor volume: How did the author define the longitudinal and transverse direction for the tumor? The authors should consider showing an example of volume assessment using B-mode ultrasound images.
  14. Line 190-191: Was the threshold for high AP or VE regions empirical? What is the threshold used in literature to define high AP or VE?
  15. Authors should add ultrasound imaging parameters like frequency, focal depth, f-number, etc.

Author Response

Please see the reviewer's comments and reply in attached file

Reviewer 2 Report

The manuscript describes immunohistochemistry parameters  and emerging ultrasound parameters (SWE) and dynamic Contrast Enhanced Ultrasound (CEUS) and secondly to study the link between the different parameters. Five different groups of 9 to 10 BALB/c female mice with subcutaneous CT26 tumors were imaged using B-mode morphological imaging, SWE and CEUS at different dates. The study could help to better explore the effect of therapies on tumor evolution, by studying the evolution of the parameters over time and by studying their correlations. Nevertheless, the small database size makes difficult to validates these results. If D7 (n=1) is not considered, it should not be included in the figures. Moreover, it should be very useful to have a more detailed explanation about it is not possible to find correlation between US parameters or between volume and US parameters or significant correlation between Immunohistological and US parameters, thus immunohistological parameters and the  SWE parameters. 

Author Response

(The authors gave the same response as above.)

Reviewer 3 Report

Comparing different diagnostic parameters using a longitudinal model is of importance in cancer research and its clinical applications. This “watching evolution in action experiments” will likely provide new dynamic information about the general tread of cancer progression, which is vital for diagnosis as well. Such types of research should be strongly encouraged.  

I appreciate the authors’ effort “to compare the sensitivity to modifications occurring during tumor growth of reference parameters (tumor volume and immunohistochemistry) and emerging ultrasound parameters (Shear Wave Elastography (SWE), and dynamic Contrast Enhanced Ultrasound (CEUS)) and to study the link between the different parameters”. Since the mouse model is much more homogenous than clinic samples (with the same amount of the type of cancer cells), one might expect to get straightforward results that support these established diagnostic platforms. However, the mixed results likely reflect the heterogeneity of tumors’ evolution, an evidence that cannot be ignored. 

While some observations can be further validated by using different cancer models, which represent different stages of tumor progression this manuscript is of interest as the platform is powerful to evaluate different diagnostic methods. For example, it is necessary to confirm the finding “that structural, immunohistochemical parameters showed little or no significant evolution over time”. This type of information should be examined in the clinic, in the context of tumor types, stages of the tumor progression. It is also possible that the details of some interpretations can be discussed or even debated, this message should be welcomed by the cancer research community. 

I thus recommend that this manuscript be accepted. 

Author Response

Please see in the attached file the reviewer's comments and reply 

Round 2

Reviewer 1 Report

  1. Figure 1: axis and labels are missing.

This manuscript is a resubmission of an earlier submission. The following is a list of the peer review reports and author responses from that submission.

Round 1

Reviewer 1 Report

General evaluation: 

In this study, author evaluated tumor evolution, including the number of T lymphocytes per mm2 and its standard deviation, the percentage area of apoptosis and the colocalization of apoptosis and vascular endothelium by using ultrasound parameters (Shear Wave Elastography (SWE), and dynamic Contrast Enhanced Ultrasound (CEUS)) in colorectal carcinoma CT26 cells xenograft mouse model. By analysis their results of parameters in immunohistological, CEUS and SWE which showed partial correlation in tumor progression. They concluded that this could help to better explore the effect of therapies on tumor evolution, by studying the evolution of the parameters over time and by studying their correlations.

Significance:

Ultrasound Elastography can estimate mechanical properties of tissue by the measurement of their elasticity. Shear Wave Elastography (SWE) provides a quantitative evaluation based on the local speed of propagation of a shear wave produced by the probe. Dynamic Contrast Enhanced Ultrasound (CEUS) uses intravenously injected microbubbles as a contrast agent. This technique can help to characterize the state of the 85 vascular network in the tumor. These two diagnostic approaches have been used in clinical cancer diagnosis in several tissues and cancer type including HCC (Kudo et al. J Medical Ultrasound 2008;16:130–9., Claudon et al. European Journal of Ultrasound. 2013; 34:11-29.) and breast cancer (Balleyguier et al. Eur J Radiol. 2009;69:14-23., Jung et al. Utrasonography. 2018;37(1):55-62). In this study, authors only tested tumor growth without any treatments by using some parameters of SWE and CEUS. However, these data did not show any association or significance in evaluating non-invasive tumor. In term of scientific significance, this study showed rough data that can not provide further suggestions in clinical diagnosis of cancer development. Therefore, there is no scientific or clinical significance presented in this study.   

Detailed critiques: 

In material and methods section

The authors used the TRITC–labeling anti-caspases 3- antibody together with immunofluorescence imaging for detecting apoptosis in tumor samples. However, the pan-caspases 3 is not able suitable for apoptosis marker. They should utilize the florescence labeling antibody against to cleaved-caspase-3 for monitoring apoptosis. Besides, the information regarding commercial supplier and catalogue or clone number is missing. 

In results section:

In figure 2, please describe the rational of testing following events, Mean Transit Time (MTT), Area Under the 244 Curve (AUC), Peak Enhancement (PE), Wash In Rate (WIR), Wash Out Rate (WOR) and Time to Peak (TTP)), SWE pa-245 rameters, histological parameters (Percent area occupied by nuclei (NU), the Number of T Lymphocytes per mm² (NTL), 246 the percent area of Apoptosis (AP) and percent area of Vascular Endothelium (VE) and co-localization of high VE and AP 247 (VE ∩ AP)). Is these parameters associated with tumor growth?

Please describe figure 3 in results section rather than in discussion section. Seems that this data is critical part in this study. 

In figures section:

In figure 3, bar chart of (VE ∩ AP), “Tumor” not “Tum or”. The space between words should be eliminated.

In Conclusion section:

Please describe your findings or contributions briefly. 

Reviewer 2 Report

The authors propose immunohistochemistry parameters and emerging ultrasound parameters (Shear Wave Elastography (SWE), and dynamic 16 Contrast Enhanced Ultrasound (CEUS) for a better non-invasively characterize the tumor microenvironment in order to reveal evidence of early tumor response to therapy and to better under-13 stand therapeutic response. The manuscript is well organized. However, there are some issues that need clarification and improvement.

Tumor volume was significantly different between "different days", but it should be interesting to show the daily evolution and quantify the progression in shorter window in larger detail. Moreover, it should interesting to clarify the explanation of the evolution of some parameters, such as the mean and standard deviation of immunohistological parameters representing the density of biological structures for the colocalization of Apoptosis and Vascular Endothelium between the day 10 and the day 14 (Fig. 1c). Furthermore, the correlations were computed using the data from all days, excluding data rom day 7, authors should explain the reason t why this date was removed. Moreover, paper needs revision of writing, brackets (e.g. in line 17), references, etc.

Reviewer 3 Report

The authors compared the volume and histological parameters with ultrasound SWE and CEUS parameters of colorectal carcinoma in a mouse model. They imaged the tumor at 4 different time points. The work is very incremental. The innovation was not clear as the relationship between SWE and CEUS parameters versus histology in the tumor was investigated previously. The authors did not find the statistical difference or correlation in most of the parameters with tumor growth. However, there was not a detailed explanation for that except selective citing of the previous literature. Was statistical difference not significant with tumor growth due to the smaller number of samples or the tumor model did not reflect tumor grown in humans? The authors need to explain their results with respect to SWE and CEUS parameter changes in humans. The authors also need to perform additional histological analysis. The manuscript can be improved by considering the following comments:

  1. Innovation is not clear. The investigation of the relationship between Histochemistry versus SWE parameters and versus CEUS parameters was done previously in both human or mouse models (example, Ref. 27 and Payen et al. UMB, 2015 Aug;41(8):2202-11). The authors need to clear the innovation in the introduction by citing previous works related to the relationship between Histochemistry versus US parameters and how their work is different from the previous work.
  2. The authors need to calculate collagen fiber concentration and correlate it with elasticity.
  3. The authors need to provide all parameter values in a table, although the parameters are not statistically different. The readers may be interested to see the parameter range with tumor growth.
  4. The number of samples was different between imaging parameters versus histological parameters. Did the discrepancy in the sample number impact the statistical analysis?
  5. Figure 3 should be added in the result section with B-mode ultrasound images. The authors need to improve figure 3 with a better x-axis, y-axis, and colorbar label. The authors should start the result section with Figure 3 by describing how different images look at different time points and what are the values at each time point.
  6. Figure 1: The authors should add the value of each mouse in the box plot as the authors had a maximum of 10 mice at each time point. Only a box plot does not give a whole idea of how the data is distributed.
  7. Lines 298-299: Why was there no correlation between VE versus tumor growth?
  8. Why did the authors choose D16 and D17 with a 1-day interval instead of wait for more than 1 day?
  9. Please provide a photograph or sketch of the experimental setup.
  10. How did the authors match the histological versus imaging planes? The size of the tumor in histological planes does not match with imaging planes (Fig. 3).
  11. If the two different systems were used for CEUS versus SWE, how did the author match the imaging plane?
  12. Why did the authors exclude day 7 and have only histological analysis from only 1 mouse?
  13. Line 280: the authors' claim of lack of change in tumor elasticity over time is not true. It has been shown previously including Juge et al 9(32) that tumor stiffens as it grows.
  14. Lines 307-313: What was the biological basis for the higher standard deviation in elasticity?
  15. Lines 328-330: As VE and CEUS parameters were not correlated, the authors should determine microvessel density from the histology.
  16. Line 107: Define D9